# Walk and Run Test in Patients with Degenerative Compression of the Cervical Spinal Cord

**DOI:** 10.3390/jcm10050927

**Published:** 2021-03-01

**Authors:** Zdenek Kadanka, Zdenek Kadanka, Tomas Skutil, Eva Vlckova, Josef Bednarik

**Affiliations:** 1Department of Neurology, University Hospital, 625 00 Brno, Czech Republic; Kadanka.Zdenek@fnbrno.cz (Z.K.S.); Vlckova.Eva@fnbrno.cz (E.V.); Bednarik.Josef@fnbrno.cz (J.B.); 2Faculty of Medicine, Masaryk University, 625 00 Brno, Czech Republic; Tomas.Skutil@gmail.com; 3Central European Institute of Technology, Masaryk University, 625 00 Brno, Czech Republic

**Keywords:** degenerative cervical myelopathy, non-myelopathic degenerative cervical cord compression, cervical spinal cord compression, 10-m walk rest, 10-m run test

## Abstract

Impaired gait is one of the cardinal symptoms of degenerative cervical myelopathy (DCM) and frequently its initial presentation. Quantitative gait analysis is therefore a promising objective tool in the disclosure of early cervical cord impairment in patients with degenerative cervical compression. The aim of this cross-sectional observational cohort study was to verify whether an objective and easily-used walk and run test is capable of detecting early gait impairment in a practical proportion of non-myelopathic degenerative cervical cord compression (NMDCC) patients and of revealing any correlation with severity of disability in DCM. The study group consisted of 45 DCM patients (median age 58 years), 126 NMDCC subjects (59 years), and 100 healthy controls (HC) (55.5 years), all of whom performed a standardized 10-m walk and run test. Walking/running time/velocity, number of steps and cadence of walking/running were recorded; analysis disclosed abnormalities in 66.7% of NMDCC subjects. The DCM group exhibited significantly more pronounced abnormalities in all walk/run parameters when compared with the NMDCC group. These were apparent in 84.4% of the DCM group and correlated closely with disability as quantified by the modified Japanese Orthopaedic Association scale. A standardized 10-m walk/run test has the capacity to disclose locomotion abnormalities in NMDCC subjects who lack other clear myelopathic signs and may provide a means of classifying DCM patients according to their degree of disability.

## 1. Introduction

Degenerative cervical myelopathy (DCM) is a neurological condition resulting from spinal cord compression arising out of degenerative narrowing of the cervical spinal canal. It constitutes the leading cause of spinal cord dysfunction in adults worldwide [1,2]. Pathological changes include osteophytosis, intervertebral disc bulging, and ligament ossification and hypertrophy, all leading to static and dynamic injury to the spinal cord [3,4]. Early diagnosis and management of DCM are vital to the provision of appropriate care for those living with this condition. Accurate diagnosis requires agreement between clinical and imaging findings. When DCM is suspected, a detailed history and physical examination should be undertaken first [2]. Common presenting symptoms include: numb and/or clumsy hand(s), bilateral arm pain and/or paresthesias, gait disturbance, Lhermitte’s sign, and urinary urgency, frequency, and/or incontinence. Objective physical signs of myelopathy include upper motor neuron signs in the upper and/or lower limbs (for example, hyper-reflexia/clonus, pyramidal Hoffmann’s, Trömner’s or Babinski’s signs, spasticity or spastic paresis of any of the extremities—most frequently spastic lower paraparesis), flaccid paresis of one or both upper extremities, atrophy of intrinsic hand muscles, sensory involvement in various distributions in upper or lower extremities, and gait ataxia with positive Romberg sign [5,6,7,8,9].

Some of the objective signs of myelopathy required for the diagnosis of DCM, detected in the course of a detailed, although largely qualitative clinical neurological examination, may serve as comparatively late indicators of cervical cord impairment. Further, degenerative compression of the cervical cord may remain free of any of the symptoms or signs of DCM. This condition–known as “presymptomatic” or “non-myelopathic” degenerative cervical cord compression (NMDCC) is highly prevalent in those above 60 years of age, involving, on average, about 40% of this European/American subpopulation [10,11]. This lies in striking contrast to the prevalence of DCM, estimated at the far lower figure of 2.3% [10]. Quantitative electrophysiological and MRI methods, however, serve to document functional or microstructural impairment in NMDCC patients, indicating that myelopathy precedes the occurrence of commonly detected clinical signs and symptoms. Thus, a diagnosis of DCM based on standard clinical bedside examination may be too late for adequate proactive treatment to be undertaken [3,12,13].

Impaired gait is one of the cardinal symptoms of DCM. Therefore quantitative gait assessment shows promise as an accurate and objective tool in the diagnosis and classification of DCM, with considerable potential in the evaluation of the impacts of therapeutic interventions [14]. Studies utilizing objective gait assessment have largely concentrated upon comparing the gait parameters of healthy individuals with DCM patients, or analyzing the pre-operative status and post-operative outcomes in DCM patients [5,15,16]. Gait impairment has been reported as a strong indication for surgical intervention and may be used as an index in the assessment of post-operative recovery [17]. Certain studies have concluded that a subtle gait disturbance is the most common and the earliest presentation of DCM [5,18,19]. Promoting this somewhat vague observation to the realm of objective assessment thus has the potential to detect early impairment and facilitate timely diagnosis of DCM [14]. There is evidence that individuals with moderate and severe DCM demonstrate slower gait speed and reduced cadence [15,20]. Correlation between quantitative assessment of gait by means of sophisticated spatiotemporal gait parameters and the degree of disability quantified by the modified Japanese Orthopaedic Association (mJOA) score has also been reported [14]. The aim of this study was, therefore, to verify whether an objective and easy-to-use gait analysis employing a standardized 10-m walk and run test is capable of detecting early gait impairment in a practical proportion of NMDCC subjects and reflecting the severity of disability in DCM patients.

## 2. Materials and Methods

### 2.1. Design

Single center, cross-sectional observational cohort study.

### 2.2. Participants

The study sample consisted of three groups: a group of DCM patients, subjects with NMDCC and a control group of healthy volunteers.

All subjects met the following inclusion criteria:
age ≥18 years;ability to walk at least 10 m without the assistance of another person.Patients or subjects were excluded if they were affected by any of the following: severe respiratory or cardiac disease hindering walking abilities or safe mobilization;history of any other neurological disorders with persistent deficit;symptomatic musculoskeletal problems affecting gait, especially coxarthrosis or gonarthrosis;symptomatic lumbar spinal stenosis (MRI of the lumbar spine performed only in patients with symptoms or signs suspected of lumbar spinal stenosis);previous surgical decompression to alleviate DCM.

Approval was granted by the local ethics committee and informed written consent was obtained from all study participants.

DCM patients and NMDCC subjects were recruited from subjects referred between January 2018 and December 2020 to a large tertiary university hospital with a multi-disciplinary center specializing in degenerative compressive neurological syndromes.

DCM patients were considered as those exhibiting generally-accepted clinical and imaging diagnostic criteria for DCM, based on the presence of at least one clinical sign and one clinical symptom of myelopathy revealed by magnetic resonance imaging (MRI) signs of degenerative discogenic and/or spondylogenic cervical spinal cord compression [5,21]. The following symptoms and signs were considered as markers of DCM.

Symptoms: gait disturbance; numb and/or clumsy hands; Lhermitte’s sign; bilateral arm paresthesias; weakness of lower or upper extremities; urinary urgency or incontinence.

Signs: corticospinal tract signs: hyperreflexia/clonus; spasticity; pyramidal signs (Babinski’s, Trömner’s or Hoffmann’s signs); spastic paresis of any of the extremities (most frequently, lower limb spastic paraparesis); flaccid paresis of one or both upper extremities; atrophy of the hand muscles; sensory involvement in various distributions in the upper or lower extremities; gait ataxia.

NMDCC patients were considered as those with MRI signs of cervical cord compression and may have exhibited one clinical myelopathic symptom, but it was essential that they were free of clinical myelopathic signs and/or lacked the combination of one clinical symptom and one clinical sign of symptomatic myelopathy required for a diagnosis of DCM.

### 2.3. MRI Examination and Assessment of Cervical Cord Compression

All subjects underwent examination of the cervical spine provided by a 1.5 Tesla MRI device with a 16-channel head and neck coil. The standardized imaging protocol included conventional pulse sequences in sagittal-T1, -T2 and STIR (short-tau inversion recovery) and axial planes (gradient-echo T2). The clinical status of all patients was blinded to the neuroradiologists who examined the cervical spine MRIs. The imaging criterion for cervical cord compression was defined as a change in spinal cord contour at the level of an intervertebral disc on axial or sagittal MRI scan compared with that at the midpoint levels of neighboring vertebrae [11,12,22].

The control group was made up of healthy volunteers without symptomatic lower limb injuries, neurological disorders, or cardiovascular or respiratory impairment that would hinder gait analysis. All volunteers underwent MRI examination of the cervical spine (either as participants in another epidemiological study or for cervical pain or cervical radiculopathy) that disclosed neither signs of degenerative cervical cord compression nor any cervical cord abnormality [11].

### 2.4. mJOA Score

The degree of disability in DCM patients was assessed in terms of mJOA score, a generally accepted disability scale. This is an investigator-administered tool used to evaluate neurological function in patients with DCM [23]. It is defined on an 18-point scale that addresses upper (5 points) and lower extremities (7 points, JOA–LE) motor function, sensation (3 points) and micturition (3 points).

### 2.5. Gait Assessment

Gait assessment was performed in standardized fashion for all participants. After a back-and-forth warming-up walk, each subject was asked to walk a 10-m walkway from a standing start, following the instructions: “Once you are given the instruction to start, you should walk as quickly as possible until you are asked to stop. You are not allowed to run”. At least one foot per step had always to make contact with the ground in order for the process to be considered “walking” [24]. Distance was calculated using markings on the track. Next, they were asked to run the same 10-m walkway as fast as they could, if possible. For patients who exhibited unstable gait, the supervision of another person was provided to prevent a possible fall. In the case of serious risk of falling, we omitted the running test. The times taken for the walk/run and the number of steps were counted by an observer and expressed as walking/running time(s), velocity (cm/s), number of steps and cadence (steps/min). No videorecording was performed.

### 2.6. Statistics

Continuous parameters were summarized as mean (X) ± standard deviation (SD) and/or median (minimum-maximum), or 5th–95th percentiles. Categorical parameters were expressed as absolute and relative frequencies. The normal distribution of continuous variables was investigated by means of graphic tools, the Kolmogorov–Smirnov and the Shapiro–Wilk tests. For assessment of correlation between gait/run parameters and mJOA and mJOA–LE scales in DCM and between gait/run parameters and age in healthy controls, the Spearman’s rank sum correlation coefficient and/or the chi-square test were deployed. Differences between the sexes in HC in gait/run parameters were calculated via the Mann–Whitney U test, while differences in gait/run parameters between groups (HC, NMDCC and DCM) were calculated via the Kruskal–Wallis and post-hoc tests with Bonferroni’s correction.

## 3. Results

### 3.1. Participant Demography

There were 100 healthy volunteers, aged 56.1 ± 13.1 (x ± SD); 55.5 (median); 30–82 (minimum-maximum) years; 52 (52%) were women. The NMDCC group consisted of 126 patients, aged 58.2 ± 9.9; 59; 30–79 years; 65 (51.6%) women. The mJOA score reached 18 points in all healthy volunteers and in vast majority of NMDCC subjects. Slight abnormality of mJOA at the level of 17 points was found in 13 out of 126 NMDCC subjects (10.3%) due to mild lack of stability and/or mild difficulties in attempt to button the shirt. No NMDCC subject had mJOA < 17. Some of them had signs of cervical radiculopathy but in all these 13 NMDCC subjects we found no clear myelopathic signs during routine clinical evaluation including those with subjective gait problems. The DCM group was made up of 45 patients, aged 59.3 ± 11.8; 58; 36–82 years, 20 (45.5%) women. There were no significant differences between the three groups in terms of age or sex proportions (*p* > 0.05). All healthy volunteers and NMDCC subjects were able to perform the 10-m walk and run test, while eleven participants from the DCM group were unable to run and took only the walk test.

### 3.2. Gait Analysis

#### 3.2.1. Healthy Controls

The values of all parameters displayed normal Gaussian distribution. All parameters correlated highly significantly with age (higher figures with advancing age for time and number of steps, lower values for velocity and cadence for both the walk and the run). They differed between the sexes (higher values of time and number of steps for both walk and run in women, no difference in cadence) (Table 1). Thus, all parameters were assessed independently in four subgroups of healthy controls (men and women aged > 60 and ≤60 years of age) and normal limits were expressed as x + 2SD (time, number of steps) or x-2SD (velocity, cadence). As the values of all the parameters obtained in both groups of patients were distributed non-normally, the 5th and 95th percentiles of values in the HC group were calculated as alternative normal limits (Table 2A).

#### 3.2.2. NMDCC

Summaries of gait parameters in NMDCC and DCM patients appear in Table 3 and Figure 1a–c. Significant differences were evident in all gait parameters among all the groups studied (*p* < 0.001; Table 3). In comparison with healthy controls (Table 3), NMDCC patients took longer to complete the ten meters at a run or walking, moved at lower speeds and required higher numbers of steps. Abnormality within the walking parameters appeared in 46.8% of NMDCC subjects. Time/velocity exhibited the highest sensitivity (45.2%), followed by number of steps (16.7%), and cadence (4.8%). All these abnormalities were disclosed in the course of investigation of time and number of steps (Table 2B).

Similarly, abnormality within the run parameters appeared in 57.1% of subjects, with the highest sensitivity exhibited by time/velocity (42.1%), followed by number of steps (32.5%) and cadence (19.0%). Again, all abnormalities were disclosed in the course of investigation of time/velocity and number of steps (Table 2B).

Abnormality of walk and/or run test parameters appeared in 66.7% of NMDCC patients (Table 2B).

#### 3.2.3. DCM

DCM patients exhibited significantly longer times/lower velocities, higher numbers of steps and lower cadence during both the walk and run tests in comparison with both healthy controls and NMDCC patients (Table 3, Figure 1a–c). Abnormality of walk parameters appeared in 71.1% of DCM patients, with the highest sensitivity for time/velocity (68.9%), followed by number of steps (31.1%) and cadence (11.1%) All abnormalities were disclosed in the course of investigation of time and number of steps (Table 2B). Similarly, abnormality of run parameters appeared in 79.4% of subjects, with the highest sensitivity for time/velocity (67.6%), followed by number of steps (64.7%) and cadence (23.5%). Again, all abnormalities were disclosed in the course of investigation of time/velocity and number of steps (Table 2B). Abnormality of walk and/or run test parameters appeared in 84.4% of DCM patients (Table 2B).

Time/velocity and number of steps as assessed from walk and run tests correlated significantly with both mJOA and mJOA–LE scales (Table 4). In addition, cadence of walk correlated with both mJOA and mJOA–LE scores, although this did not hold true for running (Table 4).

## 4. Discussion

This is, to the best of our knowledge, the first study to show that gait analysis utilizing a standardized and simple 10-m walk and run test reflects gait impairment not only in DCM patients, but in a substantial proportion (66.7%) of individuals with NMDCC. Gait impairment constitutes the most prominent clinical manifestation of cervical myelopathy, and thus its amelioration may have a substantial impact on the recovery of patient functionality [25,26].

In routine clinical practice, observational gait analysis is by far the most commonly used approach to evaluating gait disturbance in DCM, including mJOA score. The accuracy and consistency of essentially subjective observation are however, questionable, particularly for subtle gait changes [27]. Timed walk tests are more sensitive to change and are known to be valid and reliable in DCM [28], but they provide no information concerning the underlying gait parameters that have contributed to the measured speed [29]. Recently, there has been a resurgence of research interest in applying quantitative and objective gait analysis to the evaluation of patients with DCM [25,26]. Gait analysis is now largely mostly performed on the basis of a specific movement protocol that includes evaluation of the range of motion of the lower extremities, of muscle strength, and of balance differences [15,25]. An assessment may also be obtained from three-dimensional computer analysis, including a number of spatiotemporal kinetic and kinematic parameters, all of which have been demonstrated as impaired in DCM patients [26,30]. Kalsi-Ryan et al. recently presented a study that found significant differences between control subjects and patients with mild, moderate, and severe DCM, and characterized specific differences in gait parameters between severity subtypes of DCM [14]. These computer analyses, however, are hardly practical in the context of clinical neurological practice. Thus, this study was based on finding an easy and reliable test, readily available to the clinical neurologist. The protocol employed was simple and easy to reproduce, based on the straightforward instruction “walk as fast as possible, but do not run”, and followed by a run test (if possible). This contrasts with other protocols in which the walk has been undertaken at a subject selected pace.

The rationale to evaluate both walking and running abilities in degenerative cervical cord compression subjects is based on the fact that walking and running are generally considered as distinct gait modes, with strikingly different mechanics and energetics. Having the ability to walk does not mean that the individual has the ability to run, as running requires greater balance, muscle strength and greater joint range of movement [31,32]. As expected, 11 out of 45 DCM patients (24.4%) of DCM patients were not able to run, but running test disclosed abnormality in an additional 13% of DCM patients (and in 19.9% of NMDCC subjects) with normal walking test, justifying thus the usefulness of its use.

This study confirms that gait analysis based on a clinically practical and easily administered test is a highly sensitive approach to the disclosure of gait disturbance in DCM patients. The results were in close correlation, especially in terms of walking and running time and the number of steps taken, with the mJOA scale and mJOA–LE, its subscale for the lower extremities, the most widely-employed subjective scale for grading severity of disability. Abnormalities in gait parameters, however, were also found in a substantial proportion of NMDCC patients; further, this cohort exhibited significant differences in all the parameters assessed when compared with age-adjusted healthy controls. A number of reasons for these findings may be suggested. Firstly, DCM diagnosis is based on the presence of clinical symptoms and signs (at least one) of myelopathy, although some patients may complain of a certain degree of gait disturbance in the absence of clear, objective, physical signs of myelopathy [5,6]. In the light of current criteria, a diagnosis of DCM is critically dependent on the clinical expertise of the examining specialist; an objective approach to gait assessment may well serve as an additional clinical tool, enabling timely and reproducible establishment of a DCM diagnosis. Secondly, the approach employed herein based its test protocol of gait analysis on a fast walk and a run where feasible, rather than the usual assessment of a slow walk. The results arising out of a fast walk may be more sensitive than those of a “regular” walk. Of course, a run test is not suitable for DCM patients with moderate-to-severe disability. Nevertheless, in that part of the cohort herein capable of independent locomotion, 75.6% of DCM patients and 100% of those with NMDCC proved able to run, and the running test disclosed additional abnormalities in a quarter (25.5%) of them. Among the parameters assessed, not surprisingly, walking and running times showed the highest sensitivity, followed by number of steps, while cadence of walk/run did not disclose any abnormalities in patients returning normal times and numbers of steps and did not prove immediately useful. Thirdly, the parameters of walk and run correlated closely with age and sex, and therefore normal limits were adjusted for these two demographic parameters. This might have enhanced the sensitivity of the test.

Early recognition and treatment of DCM, before the onset of spinal cord damage, is essential for optimal outcomes. Unfortunately, despite the lack of any study showing a benefit of a prophylactic surgical decompression in NMDCC, some spondylosurgeons recommend and perform such intervention. Recommendations based on expert opinion and longitudinal studies on natural course of NMDCC and risk factors for progression to DCM [12,22,33] generally recommend consideration of surgical treatment in those patients who present with clinical or electrophysiological evidence of cervical radicular dysfunction or central conduction deficits disclosed by electrophysiological examination and are thus at higher risk for developing myelopathy [34,35]. There is also no clear agreement on the conservative treatment of both NMDCC and mild DCM patients. Intermittent immobilization in a cervical collar and “low-risk” activity modification together with close observation of both mild DCM patients and NMDCC subjects with high risk for progression into symptomatic DCM are usually recommended.

### Limitations of the Study

Despite the use of age and sex-adjusted normal values and the exclusion of subjects with known tandem lumbar spinal stenosis or musculoskeletal comorbidities that might have interfered with gait, a higher tendency towards degenerative changes in the lumbar spine or hip joints in patients with degenerative cervical cord compression is to be anticipated [36]. This may lead to results indicating more severe impairment in a performance-oriented test of this nature. Moreover, such a test is prone to be influenced by the motivation of the subject tested. Exclusion of patients with symptomatic lumbar spinal stenosis or musculoskeletal comorbidities that are quite frequent in older population and especially in DCM patients eliminates significant proportion of DCM patients in particular and decreases external validity of the test. Our study was performed in the Caucasian (European) population with very low prevalence of the ossification of the posterior longitudinal ligament and the results thus may be of limited value in evaluation of other populations of patients with degenerative cervical cord compression. The methodology to measure the times taken for the walk/run and to count the number of steps manually by an observer is easy to implement in the clinical setting, but might hypothetically serve as a potential source of error.

## 5. Conclusions

In conclusion, the main benefit of a standardized 10-m walk/run test in comparison to already used scoring systems, such as mJOA score, is its objective and quantitative character and sensitivity to mild gait impairment due to myelopathy. It has the capacity to disclose locomotor abnormalities in the early stages of degenerative cervical cord compression that may be confirmed as another risk factor for progression into symptomatic DCM in future longitudinal studies. Furthermore, it may support clinical diagnosis of DCM in case of vague clinical myelopathic symptoms and signs and could be employed in routine clinical practice as a tool to evaluate clinical course or effect of therapy in already diagnosed DCM.

## Figures and Tables

**Figure 1 jcm-10-00927-f001:**
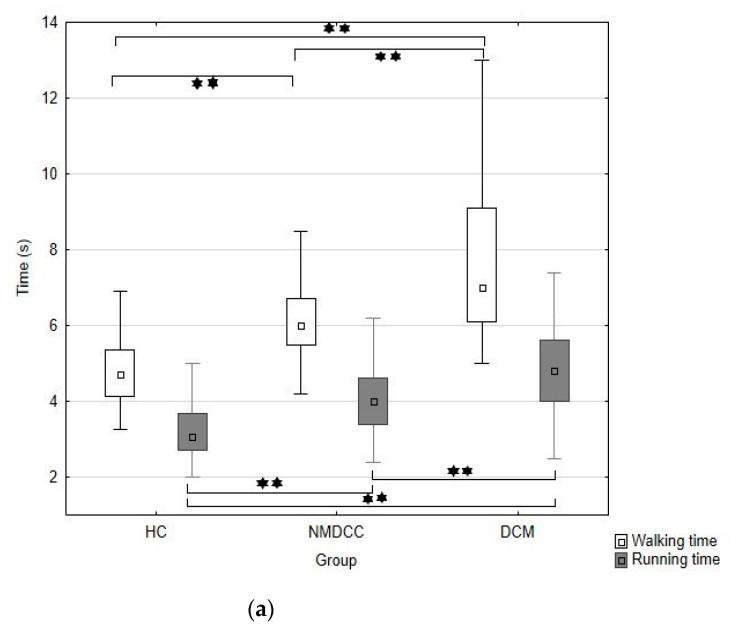
Box-plots and whisker-plots expressing median, lower and upper quartiles, minimum and maximum (without outliers) of walking/running time (**a**), number of steps taken during walk and run (**b**) and cadence of walk and run (**c**) in healthy controls (HC), non-myelopathic degenerative cervical compression (NMDCC) patients and those with degenerative cervical myelopathy (DCM). * *p* < 0.05; ** *p* < 0.01.

**Table 1 jcm-10-00927-t001:** Correlation of walk/run parameters with age and sex in healthy controls.

HC (*N* = 100)		Correlation with Age: Spearman’s Rank Correlation Coefficient: r (*p*)	Comparison between Sexes: Chi-Square Test: *p*
		Age	Sex
Time/Velocity (cm/s)	Walk	0.610/−0.610 (<0.001)	0.006 ^‡^
	Run	0.657/−0.657 (<0.001)	0.001 ^‡^
Number of steps	Walk	0.497 (<0.001)	<0.001 ^‡^
	Run	0.353 (<0.001)	<0.001 ^‡^
Cadence (steps/min)	Walk	−0.268 (0.007)	0.659 ^†^
	Run	−0.564 (<0.001)	0.707 ^†^

HC: Healthy controls; ^‡^ Significantly higher values in women; ^†^ Insignificantly lower values in women.

**Table 2 jcm-10-00927-t002a:** 10-m walking/running test: age- and sex- stratified normal limits (set in the group of healthy controls).

**Healthy Controls (*N* = 100): Subgroups**	**Parameters: 10 m Walk**			
	**Time (s)/** **Velocity (cm/s)**		**Number of Steps/** **Cadence (Steps/min)**	
	**X ± SD**	**Normal limits** **Time: X+2SD/95.perc.** **Velocity: X-2SD/5.perc.**	**X ± SD**	**Normal Limits** **N.steps: X+2SD/95.perc.** **Cadence: X-2SD/5.perc.**
Men ≤ 60 years *N* = 27	4.2 ± 0.5/ 238.3 ± 30.9	5.2/5.3 176.5/186.9	10.7 ± 1.2 153.6 ± 22.8	13.1/13.0 108.0/125.2
Men > 60 years *N* = 21	5.0 ± 0.8/ 198.8 ± 37.3	6.6/6.4 124.2/145.0	12.8 ± 2.1 158.3 ± 32.3	17.0/16.0 93.7/103.9
Women ≤ 60 years *N* = 31	4.5 ± 0.6/ 221.3 ± 28.6	5.7/5.6 164.1/178.5	12.2 ± 1.3 162.3 ± 25.1	14.8/14.5 112.1/129.4
Women > 60 years *N* = 21	6.1 ± 1.0/ 165.8 ± 30.2	8.1/8.1 105.4/110.0	14.6 ± 2.4 143.8 ± 24.4	19.4/18.0 95.0/101.5
**Healthy Controls (*N* = 100): Subgroups**	**Parameters: 10 m Run**			
	**Time (s)/** **Velocity (cm/s)**		**Number of Steps/** **Cadence (Steps/min)**	
	**X ± SD**	**Normal Limits** **Time: X+2SD/95.perc.** **Velocity: X-2SD/5.perc.**	**X ± SD**	**Normal Limits** **N.steps: X+2SD/95.perc.** **Cadence: X-2SD/5.perc.**
Men ≤ 60 years *N* = 27	2.6 ± 0.3/ 383.7 ± 58.7	3.2/3.3 266.3/304.0	8.7 ± 1.3 199.1 ± 27.6	11.3/11.0 143.9/151.3
Men > 60 years *N* = 21	3.4 ± 0.7/ 296.8 ± 55.8	4.8/4.2 185.2/237.0	9.4 ± 1.0 167.9 ± 29.5	11.4/11.2 108.9/116.9
Women ≤ 60 years *N* = 31	3.0 ± 0.4/ 336.2 ± 40.2	3.8/3.6 255.8/279.0	9.7 ± 1.2 193.6 ± 21.2	12.1/12.0 151.2/158.4
Women > 60 years *N* = 21	4.2 ± 1.0/ 238.2 ± 51.0	6.2/6.3 136.2/158.0	10.6 ± 1.0 155.9 ± 31.2	12.6/12.0 93.5/96.8

X: mean; SD: standard deviation; Perc.: percentile; N: Number.

**Table jcm-10-00927-t002b:** 

**10 m Walk—Number (Proportion) of Abnormal Values ^&^**			
Group	NMDCC (*N* = 126)	DCM (*N* = 45)	Comparison of the groups: chi-square test (*p*)
Parameter			
Time	57 (45.2%)/60 (47.6%)	31 (68.9%)/32 (71.1%)	0.006/0.007
Velocity	57 (45.2%)/60 (47.6%)	31 (68.9%)/32 (71.1%)	0.006/0.007
Number of steps	21 (16.7%)/22 (17.5%)	14 (31.1%)/23 (51.1%)	0.04/<0.001
Cadence	6 (4.8%)/33 (26.2%)	5 (11.1%)/20 (44.4%)	0.136/0.02
Any abnormality (walk)	59 (46.8%)/66 (52.4%)	32 (71.1%)/34 (75.5%)	0.005/0.007
**10 m Run—Number (Proportion) of Abnormal Values ^&^**			
Group	NMDCC (*N* = 126)	DCM (*N* = 34) ^#^	Comparison of the groups: chi-square test (*p*)
Parameter			
Time	53 (42.1%)/59 (46.8%)	23 (67.6%)/24 (70.6%)	0.008/0.014
Velocity	53 (42.1%)/59 (46.8%)	23 (67.6%)/24 (70.6%)	0.008/0.014
Number of steps	41 (32.5%)/41 (32.5%)	22 (64.7%)/22 (64.7%)	<0.001/<0.001
Cadence	24 (19.0%)/42 (33.3%)	8 (23.5%)/15 (44.1%)	0.562/0.244
Any abnormality (run)	72 (57.1%)/82 (65.1%)	27 (79.4%)/28 (82.4%)	0.018/0.054
Any abnormality (walk and/or run)	84 (66.7%)/91 (72.2%)	38 (84.4%)/40 (88.9%)	0.024/0.023

NMDCC: Non-myelopathic degenerative cervical cord compression; DCM: Degenerative cervical myelopathy; ^&^: number (proportion) of abnormalities calculated for cut-offs set as X ± 2SD/5. or 95.perc.; ^#^: eleven DCM patients were not able to run.

**Table 3 jcm-10-00927-t003:** Summary statistics of walk/run test parameters in the groups studied.

Parameters	Groups	HC	NMDCC	DCM	Kruskal–Wallis *p* Value *
		X (SD); Median (Min.–Max.)			
Walk time (s)		4.9 (1.3); 4.7 (3.3–13.6) ^a^	6.2 (1.1); 6.0 (4.2–9.9) ^b^	7.2 (2.5); 7.0 (5.0–18.0) ^c^	<0.001
Walk velocity (cm/s)		209.0 (42.5); 212.5 (73–306) ^a^	165.9 (27.2); 167 (101–238) ^b^	139.6 (34.2); 150 (56–200) ^c^	<0.001
Walk steps (No.)		12.4 (2.2); 12 (9–23) ^a^	13.2 (1.9); 13 (8–18) ^b^	14.8 (2.9); 15 (10–23) ^c^	<0.001
Walk cadence (steps/min.)		155.2 (29.2); 152.9 (100.0–263.4) ^a^	130.7 (20.6); 130 (53.3–228.6) ^b^	120.7 (18.0); 120 (63.3–159.4) ^b^	<0.001
Run time		3.3 (0.9); 3.1 (2–8) ^a^	4.1 (0.9); 4.0 (2.4–6.7) ^b^	4.6 (1.4); 4.8 (2.5–9.4) ^c^	<0.001
Run velocity (cm/s)		320.1 (74.1); 323.5 (125–497) ^a^	255.9 (56.5); 250 (149–416) ^b^	219.0 (61.7); 221 (150–400) ^c^	<0.001
Run steps (No.)		9.6 (1.4); 10 (6–12) ^a^	11.3 (2.4); 11 (7–18) ^b^	12.8 (3.0); 13 (8–22) ^c^	<0.001
Run cadence (steps/min.)		181.8 (33.3); 182.6 (75.0–264.0) ^a^	167.1 (25.9); 169.4 (114.3–266.7) ^b^	160.2 (20.0); 161.2 (108.5–200.0) ^b^	<0.001

HC: Healthy controls; NMDCC: non-myelopathic cervical cord compression; DCM: degenerative cervical myelopathy; X: mean; SD: standard deviation; * *p*-value represents comparison of all the groups (Kruskal–Wallis test); post hoc tests: a,b,c—same letters marking values of categories within any given row denote groups that are not mutually statistically different.

**Table 4 jcm-10-00927-t004:** Correlation between severity of disability and walk/run parameters in DCM patients.

DCM Patients (*N* = 45)		Spearman’s Rank Correlation Coefficient r (*p*)	
		mJOA:	mJOA LE
Time (s)	Walk	−0.766 (<<0.001)	−0.790 (<<0.001)
	Run	−0.505 (0.002)	−0.568 (<0.001)
Velocity (cm/s)	Walk	0.766 (<<0.001)	0.790 (<<0.001)
	Run	0.505 (0.002)	0.568 (<0.001)
Number of steps	Walk	−0.589 (<0.001)	−0.649 (<<0.001)
	Run	−0.485 (0.004)	−0.471 (0.005)
Cadence (steps/min)	Walk	0.514 (<0.001)	0.483 (<0.001)
	Run	0.173 (0.329)	0.239 (0.173)

DCM: degenerative cervical myelopathy; mJOA: modified Japanese Orthopaedic Association scale; mJOA LE: modified Japanese Orthopaedic Association subscale for lower extremities; <<0.001: *p* value less than 10^−6.^

## Data Availability

The data presented in this study are available on request from the corresponding author. The data are not publicly available.

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
