# Peer review of "Walk and Run Test in Patients with Degenerative Compression of the Cervical Spinal Cord"

_jcm, 2021, doi:10.3390/jcm10050927_

Round 1

Reviewer 1 Report

Kadanka Jr. et al report a large cross-sectional study in 55 DCM patients, 126 NMDCC subjects, and 100 healthy controls investigating simple walk and run tests. The methods are sound and the study is interesting and important, as standardized methods for gait assessment in DCM are lacking. The manuscript is well written and the discussion and conclusions reasonably appropriate. The analysis is reasonable, although I think the authors could organize the results in a simpler and more direct way (e.g. tables 2A and 2b are confusing and it is unclear why the results are stratified by age above/below 60 when age is analyzed using Spearman elsewhere). In terms of age, I would suggest the authors provide scatter plots of each measure and its relationship to age, to help understand if these are linear or non-linear. Also, what was the mJOA in HCs and NMDCC? As the discussion states, some of the NMDCC subjects have mild myelopathy that did not quite meet the objective definition proposed by the authors (e.g. lack of hyperreflexia), so the mJOA data for these subjects would be very interesting. Furthermore, there is insufficient discussion of the limitations of this study - one area of great concern is the external validity due to exclusion of subjects with musculoskeletal conditions and lumbar stenosis, which probably eliminates about half of DCM patients. Also the method of counting steps was not described - was this done with a human observer and no video recording? This seems to be a potential source of error that should be included in the discussion of limitations.

Reviewer 2 Report

In this paper the authors developed a standardized 10-meter walk and run test to screen for symptomatic patients suffering from both non-myelopathic degenerative cervical cord compression (NMDCC) and degenerative cervical myelopathy (DCM). They address a highly relevant topic since clear evidence on the indication and optimal timepoint for surgery in patients suffering from cervical cord compression without myelopathy is lacking.

However, the Nurick grading system and the JOA/mJOA scores are both well-developed and easily applicable tools to screen for DCM. Thus, based on the provided data, an additional benefit of the 10m run and walking test in screening and stratifying for DCM patients cannot be seen.

Most importantly, my major concern is related to the lack of an added value in determining the surgical indication in patients with mild to no symptoms and radiological signs of cervical spinal cord compression. This is complicated since both clinical and imaging outcome/follow-up data on patients with NMDCC are not provided. Hence. the discussion section needs major revision to outline the limitations and these perspectives.

Further comments:

  • Although the authors have mentioned that patients with a history of lumbar spinal stenosis have been excluded from analysis: have tandem degenerations at the thoracolumbar levels been actively ruled out by MRI in all patients with NMDCC?
  • How were numbers calculated if patients couldn’t run in the DCM cohorts?
  • A 10 m run seems like a short distance for a running test with an increased risk of false measurements. How were the measurements performed? Laser? Manually? What was the reason to choose 10 m as a standard distance? Have other distances been tested?

Reviewer 3 Report

This is an important study to evaluate walk and run ability in the DCM patients, and compared the potential with non-myelopathic patients and healthy volunteer. In particular, the patients without myelopathic symptoms also showed significant worse ability of gait function through their conducted test. It would be suitable for publication after clarifying several questions.

  • At 3.1 in Results section, they wrote “There were no significant differences between the three groups in terms of age or sex proportions (p <0.05).” It seems like p > 0.05.

  • DCM encompasses several pathology including spondylosis, OPLL, disk herniation. Was there any difference among these conditions in demographics and outcomes of gait test?

  • The authors should clarify what is the difference of their aim to evaluate walk and run tests. Was there any functional difference between the walking and running abilities?

  • They wrote in Discussion section that “Early recognition and treatment of DCM, before the onset of spinal cord damage, is essential for optimal outcomes.” They should propose what treatment is appropriate for the patients with non-myelopathic DCC patients. Should it be a conservative treatment, and if so, what intervention? Otherwise, some doctors can perform surgical treatment for this non-myelopathic patients as a prophylactic intervention.

Round 2

Reviewer 3 Report

It is well-revised and time to consider publication.

Author Response

Thanks a lot for your positive comments of our revision and for Your recommendation to publish in this journal.